# Global Neuropsychopharmacological Prescription Trends in Adults with Schizophrenia, Clinical Correlates and Implications for Practice: A Scoping Review

**DOI:** 10.3390/brainsci14010006

**Published:** 2023-12-20

**Authors:** Jiangbo Ying, Qian Hui Chew, Yuxi Wang, Kang Sim

**Affiliations:** 1East Region, Institute of Mental Health, Singapore 539747, Singapore; 2Research Division, Institute of Mental Health, Singapore 539747, Singapore; 3West Region, Institute of Mental Health, Singapore 539747, Singapore

**Keywords:** schizophrenia, antipsychotics, antidepressants, mood stabilizers, pharmacoepidemiology

## Abstract

It is important to examine the psychotropic prescription practices in schizophrenia, as it can inform regarding changing treatment choices and related patient profiles. No recent reviews have evaluated the global neuropsychopharmacological prescription patterns in adults with schizophrenia. A systematic search of the literature published from 2002 to 2023 found 88 empirical papers pertinent to the utilization of psychotropic agents. Globally, there were wide inter-country and inter-regional variations in the prescription of psychotropic agents. Overall, over time there was an absolute increase in the prescription rate of second-generation antipsychotics (up to 50%), mood stabilizers (up to 15%), and antidepressants (up to 17%), with an observed absolute decrease in the rate of antipsychotic polypharmacy (up to 15%), use of high dose antipsychotic (up to 12% in Asia), clozapine (up to 9%) and antipsychotic long-acting injectables (up to 10%). Prescription patterns were mainly associated with specific socio-demographic (such as age), illness (such as illness duration), and treatment factors (such as adherence). Further work, including more evidence in adjunctive neuropsychopharmacological treatments, pharmaco-economic considerations, and examination of cohorts in prospective studies, can proffer insights into changing prescription trends relevant to different treatment settings and predictors of such trends for enhancement of clinical management in schizophrenia.

## 1. Introduction

Schizophrenia is a major mental illness characterized by positive symptoms, such as hallucinations and delusions, and negative symptoms, such as social withdrawal and amotivation [1]. The prevalent cases of schizophrenia globally have been on the rise, as evidenced by an increase in reported cases from 13.1 million in 1990 to 20.9 million in 2016, and this has contributed to a global disease burden of 13.4 million years of life lived with disability [2]. In the United States (US), the economic burden of schizophrenia has been reported to be more than 60 billion dollars per year [3] and in the United Kingdom (UK), the cost of schizophrenia has been estimated to be 6.7 billion pounds annually, with direct costs of 2 billion pounds on health and social care budgets [4]. Clearly, schizophrenia exerts a large burden on patients and society and needs appropriate treatment. 

The treatment of schizophrenia involves pharmacological therapy in conjunction with psychosocial interventions, such as cognitive behavioral therapy, social support, and rehabilitation [5]. Antipsychotics (APs) are the primary psychotropic medications to treat schizophrenia. Many different APs, such as haloperidol, olanzapine, risperidone, and clozapine (CLZ), have been assessed for their efficacy and safety in controlling psychotic symptoms [6,7,8]. APs are particularly effective in treating positive symptoms, but they have limited effectiveness in treating negative symptoms [9]. Over the past few years, research has suggested the pathophysiology of schizophrenia may go beyond the dopamine hypothesis and involve other molecular targets, such as glutamatergic, cholinergic, and gamma-aminobutyric acid (GABA) receptors [10,11]. Other psychotropic medications, such as mood stabilizers (MSs) and antidepressants (ADs), have been examined as adjuvants to APs in the treatment of symptoms such as aggression [12] and negative symptoms [13]. Lithium, which has been the mainstay treatment for bipolar disorder [14], acts on the GABA receptor to restore the excitatory–inhibitory neurotransmitter levels [15]. Valproate also acts on the GABA receptor and may target other signaling pathways, such as arachidonic acid cascade [16]. ADs modulate several neurotransmitter systems, including the monoaminergic system [17]. It has been reported that add-on citalopram may reduce the level of negative symptoms in schizophrenia [18]. 

Although there are treatment guidelines for schizophrenia, prescription patterns of psychotropic medications are often influenced by many other factors, such as the clinician’s own training, local prescription practices, the nature of the healthcare system, and the delivery of healthcare [19]. It has been reported that treatment guidelines may not fully reflect routine clinical practice [20]. For example, psychiatrists in Japan frequently prescribed antipsychotic polypharmacy (APP) due to their positive attitudes towards this practice [21]; and in Asia, depot APs were prescribed to improve treatment adherence in severely ill hospitalized individuals [22]. There have been cross-sectional or cohort studies that explored the prescription patterns for schizophrenia. One study in Asia found that among 3537 patients with schizophrenia, 31.3% and 80.8% received first- and second-generation APs, respectively, and 13.7% were prescribed an MS [19]. Another study assessed 2003 patients in Asia and found that 82.14% received APs and 14.7% had MS monotherapy [23]. To the best of our knowledge, there are no recent studies that have examined the global psychotropic prescription trends in schizophrenia. 

It is important to understand the psychotropic prescription patterns in schizophrenia, as it can inform regarding changing treatment choices and related patient profiles. Understanding clinical correlates and outcomes associated with different prescription trends can potentially help clinicians identify clinical profiles that may benefit from certain psychotropic medications. In addition, knowing the details of prescription practices allows healthcare professionals to balance the potential benefits of medications with their side effects and risks. Thus, the aims of the current study were to: (1) assess the global patterns of real-world psychotropic agents (APs, MSs, ADs) used in adult patients with schizophrenia spectrum disorder; and (2) evaluate the inter-relationships between psychotropic medication use and clinical correlates. This study would address the gap in existing research by providing a comprehensive and up-to-date summary of the worldwide psychotropic prescription patterns in schizophrenia. 

## 2. Materials and Methods

This study utilized the five-step framework developed by Arksey and O’Malley for scoping reviews [24] to direct the process. The first step consisted of identifying the main research questions explored in our review, which were: (1) What were the prescription patterns of psychotropic medications in adult patients with schizophrenia spectrum disorder? and (2) What were the inter-relationships between psychotropic agent use and clinical correlates?

The second step focused on the identification of relevant studies. The literature search was performed on the PubMed database for articles published between January 2002 and November 2023 by including the keyword “schizophrenia” in the title/abstract of the article together with a combination of the following keywords related to psychotropic medication (“psychotropic*”, “antipsychotic*”, “antidepressant*”, “mood stabilizer*”, “medication”, “drug*”). The keywords “pattern”, “epidemiolog*”, “pharmacoepidemiolog*”, “trend*”, “prevalence”, “frequency”, and “percentage” were used to specify the topic of interest, and the search was restricted to certain study designs using the keywords “cross-sectional”, “register”, “claim”, “cohort”, and “observational”. 

The third step involved the study selection. According to the inclusion criteria and exclusion criteria, all authors J.Y., Q.H.C., Y.X.W. and K.S. independently screened the retrieved literature. In the case of a disagreement, all authors (J.Y., Q.H.C., Y.X.W. and K.S.) would discuss until a consensus was reached. Selection criteria were as follows: (1)Original papers published in English.(2)Findings that were specific to adult patients with schizophrenia spectrum disorder.(3)Available data on psychotropic prescription patterns.

Exclusion criteria included: (1)Results are specific to a particular sub-population only (e.g., elderly, child/adolescent, pregnant women).(2)Results are specific to a certain psychotropic medication class (e.g., depot AP only, olanzapine vs. risperidone only).(3)Review papers, such as meta-analyses and systematic reviews.

The fourth and fifth steps encompassed the data collection, summarization, and reporting of findings. For each included study, we extracted variables including the characteristics of subjects, nature and aim of the study, type, and prevalence of APs, MSs, and ADs prescribed, as well as the main correlates reported. 

## 3. Results

There was a total of 88 studies included in our review published from 2002 to 2023. The PRISMA flowchart of the papers screened and included is shown in Figure 1. The main details of the included studies are shown in Appendix A. There was an average of 20,819 participants in 87 studies that reported sample sizes. The percentage of male participants ranged from 31.9 to 95 in 71 studies that reported this data. Participants were 41.6 years old on average based on 41 studies. The majority of the studies were conducted in Asia (27 studies, 30.7%), followed by the US and Canada (26 studies, 29.5%), and the UK and Europe (24 studies, 27.3%). A small number of studies were also conducted in Africa (five studies, 5.7%), Australia and New Zealand (three studies, 3.4%), Brazil (one study, 1.1%) and across regions (two studies, 2.3%).

### 3.1. Antipsychotic Prescription Trends and Clinical Correlates

APs are generally regarded as the first-line treatment for patients with schizophrenia and were given to the majority of these patients, although a sizable number of studies reported patients who received no dispensed APs. The frequency of AP use was 67–98.8% in schizophrenia patients in these studies [25,26,27,28,29,30,31,32,33,34,35,36,37,38,39]. Most patients (52.3% to 90.9%) were on AP monotherapy [28,29,30,31,34,38,40,41,42,43,44,45,46,47,48,49,50,51,52], except for five studies [27,53,54,55,56]. Monotherapy AP prescription generally showed an upward trend over the years [38], particularly for second-generation AP (SGA) monotherapy [43]. Monotherapy is generally more frequent in patients receiving SGA [29,40,47,56,57,58], particularly those initiated on olanzapine [53]. Nonetheless, there were several earlier studies reporting higher rates of first-generation AP (FGA) as compared to SGA monotherapy [31,45,50,54,59,60].

Most studies reported greater SGA use (ranging from 31.2% to 93.1%) [19,28,29,38,39,55,57,61,62,63,64,65,66,67,68,69,70] as compared to FGA (8.9% to 71.3%) [19,28,29,38,39,55,57,61,62,63,64,65,66,67,68,69,70]. Prescription rates of FGA fell over the years (by 11–26.1%) [35,38,43,49,55,57,59,68,69,70,71,72,73,74], except for one study [75]. Use of SGA increased over time (by 3.2–73.9%) [35,38,43,49,55,57,59,68,69,70,71,72,73,74], and more than doubled over seven years in one study (rates of use from 27.5% to 76.9%) [55]. 

Risperidone (16.3–44.8%) was the most prescribed oral AP in seven studies [19,27,38,39,63,67,68,76], or among the top few in other studies (12.5–69.2%) [25,31,32,45,46,61,77,78,79]. Olanzapine (13.7–91.3%) [19,25,27,32,35,39,45,61,63,77,78,79,80,81,82], quetiapine (7.6–17.8%) [25,32,35,39,79], aripiprazole (11.6–31.9%) [35,43] and haloperidol (17.9–62%) [31,32,38,51,61,68] were also frequently prescribed. Studies investigating trends in AP use reported that ziprasidone [83], olanzapine [49,78,80], and quetiapine [49,75] use increased significantly over time, while haloperidol use fell [38,43,68,78,80]. Mixed results were reported for aripiprazole [43,75] and risperidone [38,43,49,68,78,80]. 

Factors associated with the prescription of an SGA included younger age [84], and shorter duration of hospitalization [84]. Those on SGA also had a lower score on aggression regardless of dose [85]. Factors associated with the prescription of an FGA included anticholinergic use [29] and having received inpatient treatment in the past year [45]. 

#### 3.1.1. Dosing of Antipsychotic Prescribed

Approximately 80% of schizophrenia patients were on a dose of less than 600 mg chlorpromazine (CPZ) equivalents/day [19,43]. The average AP dose in studies based in Asia ranged from 273 mg to 683.5 mg CPZ equivalents/day [19,50,63,65,68,69,70,86,87,88], compared with 232 mg CPZ equivalents/day [73] in Denmark, and 467 mg CPZ equivalents/day in the US [26].

Higher mean daily doses were associated with use of high-potency agents [26], more recurrent episodes of illness [45], inpatient treatment in the past year [45], use of adjunctive MS [65], being male [26,52], younger age [26,45], being Indigenous Australians [52], and being under involuntary treatment [52]. One study found that the mean dose of APs increased over the years, with the CPZ equivalent dose/day doubling from 1996 to 2005 [73], although others reported a decrease [68,69,70,74]. 

High-dose AP prescription rates varied according to definitions. When determined as the ratio of prescribed daily dose (PDD) to defined daily dose (DDD) greater than 1.5 (PDD/DDD > 1.5), the high dose prescription rate ranged between 18.4–45.3% [75,89], and was 8.1% when defined by British National Formulary (BNF) percentage exceeding 100%, (BNF > 100%) [89], and 2.2%–17.9% when using CPZ equivalents > 1000 mg in patients with schizophrenia [49,68,87,89,90]. When defined as >1200 mg CPZ equivalents, the high-dose AP prescription rate was 4.7% [19]. One study involving a population of patients with severe and persistent schizophrenia reported an average CPZ equivalent dose of 1386.6 mg/day [78]. Of note, studies found that AP drugs were under-dosed at a rate of 26.7% [87], and excessively dosed in 0.8% of patients when compared to treatment guidelines [91]. Patients on polypharmacy with AP long-acting injectable (LAI) were more likely to be on doses above the BNF limit compared to those on oral-only APP [54]. 

Over time, the high-dose AP prescription rate decreased [49,68], with the absolute rate more than halved within Asia from 17.9% in 2001 to 6.5% in 2004 [68]. In a separate study in Korea, there was a downward trend of high-dose AP monotherapy prescription (30.4% to 18.4%) [75] but an upward trend for high-dose APP (34% to 45.3%) [75]. 

High-dose AP prescription was associated with sociodemographic factors (such as male gender), treatment factors (such as APP and inpatient treatment), and illness factors (such as first-episode psychosis and severe psychopathology). Details of the various clinical correlates with high-dose AP use are seen in Table 1.

#### 3.1.2. Antipsychotic Treatment Adherence and Discontinuation

For first-line treatment, oral APs were the preferred choice, with 71% of patients with schizophrenia who initiated treatment for the first time being given an oral AP [92]. CLZ (34%) and FGA LAI (34%) were the preferred choice for second-line or later treatments [92]. Nonetheless, adherence to AP treatment could wane, with less than half of the patients continuing their treatment beyond one year in Italy, Spain, the UK and the US [41,93], and was especially associated with APP [41], alcohol abuse/dependence [94], use of MSs [94], involuntary admission [94], prior arrests [94], greater symptom severity [94], previous AP use [94], lack of social activities [94], as well as age below 25 years [41]. In comparison to those with a more continuous pattern of AP treatment, patients who engaged in moderate or light usage had 52% or 72% increased odds of hospitalization for schizophrenia [95]. In addition, poor treatment adherence resulted in an average length of hospitalization that was 20% longer than that of those who were adherent to treatment [95]. 

Male gender [91], older age [91], and use of AD [91] were associated with lower likelihood of treatment discontinuation. The likelihood of drug switching was higher for those who were inpatients [57] and with SGA use [59]. CLZ [91,94] and olanzapine use [94] had the highest frequency of medication maintenance at 12 months while quetiapine and amisulpride had the lowest in a previous study [94]. Switching and augmentation of initial AP medication was associated with significant increases in healthcare costs compared to monotherapy [30]. The main reason cited for an AP switch/discontinuation was ineffectiveness [37,91].

#### 3.1.3. Antipsychotic Long-Acting Injectables

Most studies reported limited use of AP LAI (2% to 25%) [28,29,31,35,43,44,46,50,55,59,62,68,69,72,73,80,85,90,93,96], while a few studies reported significant use of AP LAI (35.6% to 58%) [27,45,54]. The most common AP LAI used were risperidone (38.8%) [97], zuclopenthixol (11.4–17.8%) [27,97], and fluphenazine (5.4–52.4%) [27,54,97]. Out of all schizophrenia patients on LAI, a significant proportion were on an FGA LAI (48.9–56.6%) [28,93,97], and associated with concomitant oral AP use at the one-year follow-up period [97]. 

There was an overall decrease in LAI prescription over time in Asia (2.6% to 9.5% drop) [55,68,69,73] and some Western countries [59,80]. However, this differed by type of AP LAI. There was a 4–7% increase in the proportion of SGA LAI prescriptions [92] and a 13–19% increase in the proportion of FGA LAI prescriptions from 2013 to 2017 [92] in one study in Australia, while another study reported a 7.8% drop in FGA LAI prescriptions from 2000 to 2003 [49].

Only one study explored correlates with AP LAI use in a regression model and found that those treated with FGA LAI tended to be African American and non-veterans, had a history of prior arrests and alcohol and illicit substance use, and had greater symptom severity and psychiatric hospitalization in the previous year [98]. Although SGA LAI use (21%) was lower than that of FGA LAI (34%) and CLZ (34%) in second-line or later treatments for schizophrenia patients, Pai and colleagues [92] reported that adherence to SGA LAI was better than other treatment modalities. 

#### 3.1.4. Antipsychotic Polypharmacy

Observational studies reported APP rates of approximately 3.7–57.7% in the US [30,34,41,53,71,93,99,100,101,102,103], 2.3–69.9% in Europe [27,29,31,32,35,40,45,58,73,80,93,104], 50.9% to 70.4% in Nigeria [54,77], 22.7–28.2% in Ethiopia [51,60], 67% in Uzbekistan [32], 43.9% in India [48], 19.9–44% in Japan [37,44,47,86], 8.3% to 34.2% in China and Taiwan [28,46,47,50,55,63,64], 48.3% in Korea [75], 20% in Australia [52], and 40.1–45.7% across several Asian countries [19,38,65,69,70,90,105]. 

The combination of two FGA was common in Ethiopia and Nigeria (13.5–44.4%) [51,54,60,77], although this combination decreased over the years [38,63,73,75,104], being replaced by the increased preference of two SGA in combination therapy (up to 41.7% out of all patients) [38,63,73,75,104], or a combination of FGA and SGA [38,71,73,104]. Rates of SGA combination therapy ranged from 6.4–57.7% out of the entire cohort of patients [19,38,47,48,63,73,75,78,99], while a combination of FGA and SGA was prescribed at a rate of 2.1–55.3% [19,28,29,31,38,40,45,47,48,58,60,61,63,67,71,73,75,77,84,99].

On average, each clinician had an average of 7% ± 9% of patients with APP [106], and it was observed in the same study that prescribers who managed patients with a high frequency of schizophrenia-related hospitalizations, or those with a larger proportion of Hispanic or non-Hispanic black patients, a smaller percentage of patients with disabilities, and those with a low overall volume of AP prescriptions were much less likely to be associated with APP versus prescribers with less severely ill patients [106]. The main reason cited by clinicians for an increase in the number of AP prescribed was “ineffectiveness” during clinical management [37].

APP was associated with socio-demographic factors (such as males and living alone), treatment factors (such as long-acting AP injectable use, CLZ use, inpatient setting, greater number of hospitalizations, and non-treatment adherence), and illness factors (such as longer illness duration and more psychiatric comorbidities). Details of the various clinical correlates with APP are seen in Table 2. 

In terms of changes over time, five studies on prescription trends showed that APP generally increased over the years [48,63,71,73,75,100,104], and approximately doubled in two studies [48,73]. This was particularly true for SGA APP (up to 40% increase) [38,43,63,73,75]. However, there was the suggestion of a slight decline in APP in recent years, particularly among those with chronic schizophrenia [74]. Seven studies reported a reduction of between 1.2% to 14.9% [38,47,48,49,55,69,70,99,104].

In terms of the number of APs, of note, 9.1–57.7% of patients were on two AP medications [27,29,31,38,40,46,47,48,49,60,63,75,78,92,99,104], up to 38.4% of patients were on three APs [27,29,40,46,48,60,63,75,77,78,92,99,104], and 0.12–10.1% were on four or more APs [27,29,40,63,75,78,99,104], with 1.6% on doses that exceeded the maximum dose [58]. 

In terms of doses of APP, the CPZ equivalent dosage of AP for patients on APP ranged from 380.3 to 1386.6 mg/day [50,78]. Patients on APP received lower dosages of first-generation agents compared to those on monotherapy [102]. However, patients on APP tended to receive more than twice the AP dose compared to those on AP monotherapy [73]. 

#### 3.1.5. Clozapine Use

There was a wide range of prescription rates for CLZ in both Europe and the US, ranging from 3–22.7% in Europe [25,27,29,31,32,35,40,45,58,79,80,85,107] and from 2–46.2% in the US [39,42,59,78,108]. In non-European countries, rates of CLZ use were 3.8% in Nepal [61], 66% in Uzbekistan [32], and 35% in New Zealand [49], 25% in Australia [52]. Studies across several Asian countries reported rates of 15.6–18.9% [19,38,68], although studies in specific Asian countries suggested significant inter-country differences. Rates of CLZ use in China were generally high (15.6–44%) [46,50,63,84,88], and CLZ was the most widely prescribed AP in three studies [19,46,49]. The use of CLZ in schizophrenia patients in Japan was comparatively limited, with rates of 0.2–7.1% being reported [43,44]. CLZ prescription fell significantly over time [80], by approximately 50% in one study from 18% in 1998 to 9% in 2003 [72], although four studies reported an increase in CLZ use [38,49,59,68]. CLZ use increased from admission to discharge for patients with schizophrenia [45,107].

In an earlier study, it was noted that each clinician had an average of 7% ± 10% of patients with CLZ use [106]. Prescribers were less likely to use CLZ if they had a larger proportion of Hispanic and non-Hispanic black patients, a smaller proportion of patients with disabilities, or a low volume of AP prescriptions overall [106].

Patients on CLZ had the highest frequency of medication maintenance over time (79.5–83.9%) [59,79,92,94], and had a lower discontinuation rate compared with olanzapine and risperidone [91]. In terms of CLZ dosing, patients on CLZ received a mean dose ranging from 69 to 507.7 mg/day [29,32,40,46,80,91,94,102].

### 3.2. Mood Stabilizer Prescription Trends

Adjunctive MS was prescribed at a rate of 13.6–51.6% in the US [33,41,71,78,103,109], 3.1–28.5% in Europe [29,31,35,56,79,80,85,94,96,107], and 0.9% in Nigeria [77]. Adjunctive MS use was highest in Italy when compared to European countries [79]. In Asia, the rates were 4.2–19.4% in China and Taiwan [28,46,47,55,88], 20.9% in Korea [75], and 22.1–37% in Japan [44,47,86,87]. A study across several Asian countries reported rates of 13.6–23.7% [19,65,69], with 12.8% on one MS, 0.76% on two MS, and 0.03% on three MS [65]. 

Rates specific to adjunctive lithium use were low and largely similar across countries, ranging from 2.6–7.8% in Europe [35,58,93], 1.7–6.6% in the US [93,109], and 1.6–5.1% in Asia [65,69,74,88]. Overall, the most commonly prescribed MS was valproate (4.4–16.7%) [65,69,88,109,110]. 

Prescription rates of adjunctive MS for schizophrenia patients generally increased over the years [55,69,71,75] by between 3.5% to 15.2% [55,69,71,75]. The use of valproate increased significantly over the years [69], while the use of lithium and carbamazepine declined [69,72,81]. In terms of dosing, the average MS dose was 613 ± 456 mg/day Li equivalents in a study within an Asian research consortium involving schizophrenia patients [65].

Adjunctive MS use was associated with illness factors (such as longer duration of illness, aggression, affective symptoms, and previous suicidality) and treatment factors (such as APP and AD use). Details of the various clinical correlates with adjunctive MS use are seen in Table 3. 

### 3.3. Antidepressant Prescription Trends

The prevalence of adjunctive AD use was 4.4–27.4% in China and Hong Kong [28,46,47,64,74], 47.7% in Taiwan [55], 8–42.3% in Japan [44,47,86,87], 8.5% in Korea [75], and 8.7–11.7% in several countries across Asia [19,70]. Adjunctive AD was prescribed at rates of 30.8–56.2% in the US [33,41,59,71,78,93,103], 5.4–45.7% in Europe [27,29,31,32,35,36,58,62,73,79,80,81,85,93,94,96,107], 4.4% in Nigeria [77], and 36% in Uzbekistan [32]. AD use was highest in the UK/Ireland and France across Western countries [79]. 

Prescription of adjunctive AD for schizophrenia generally exhibited an upward trend over the years [55,64,70,71,73,75,80,81,94], and nearly doubled in two studies (18.5% in 1995 to 35.6% in 1999 [71], 24.3% in 1996 to 40.6% in 2005 [73]). In terms of dosing, the mean annual DDD for AD prescriptions increased over time [73]. 

The following factors were associated with more AD use: female gender [79,96,110], younger age [70,110], less severe illness/fewer positive symptoms [64,70,79], more depressive symptoms [64,79], less use of FGA [64], more use of benzodiazepines (BZD) [64,70,110], current or prior use of an MS [79,110], use of anticholinergic/anti-Parkinsonian drugs [110], earlier age of onset of illness [64], co-occurring depression, post-traumatic stress disorder (PTSD) or substance use disorder [111], previous suicidality [110], outpatient in a tertiary/specialty treatment center [64,111], no previous psychosis episodes requiring inpatient care [110], better compliance [79], experiencing loss of libido as a side effect [79], and no history of homelessness [111]. 

### 3.4. Prescription Trends of Other Psychotropic Medications

#### 3.4.1. Anticholinergics

Anticholinergics were prescribed at a rate of 9–63.7% in patients with schizophrenia [19,27,29,31,50,55,66,73,75,80,86,90,94]. Anticholinergic use was highest in the UK/Ireland and Greece across Western countries [79]. Anticholinergic dosages were higher in those with high AP doses [86], those on APP [86], and those on FGA [86]. 

Anticholinergics were used with decreasing frequency over the years in four studies [55,73,80,94], although another study reported approximately 5% increase over five years [75], noted from admission to discharge (9.3% to 11.6%) [107]. An anticholinergic prescription was associated with patient non-compliance [79], absence of substance abuse [79], prior olanzapine or CLZ use [79], more positive symptoms [79], more extra-pyramidal side effects (EPSE) [79], high-dose AP use [86], APP [86], and use of FGA monotherapy [86]. Patients treated with a combination of LAI and oral APs [31] or on any FGA [29] were significantly more likely to be prescribed anticholinergics, versus SGA [66]. 

#### 3.4.2. Benzodiazepines

There were large variations in rates of prescription of adjunctive BZD with anxiolytic and hypnotic properties across countries. In a study surveying several countries within Asia, the average rate of BZD prescription was 27.8%. The lowest rate (8.6%) was in Hong Kong [74], while the highest (89.4%) was reported in Taiwan [55]. Rates of BZD use in other Asian countries were as follows: China (19.6–31.6%) [28,47,63,64], Korea (58.7%) [75], and Japan (49.9–68%) [44,47,87]. 

Similarly, in Europe, the average prescription rate was 26.8% [94], with the lowest reported in Sweden (6–32.8%) [25,35,36], and the highest in Austria (89.7%) [80]. Rates in other European countries were as follows: 22.3% in Finland [35], 28.1% in France [85], 40–54.8% in Spain [27,62], 36.2–54.3% in Belgium [29,107], 37% in Germany [32], 37.77% in Romania [56], and 47% in Italy [31]. Rates of use in the US saw smaller variations, ranging from 14.6–33.5% in several studies [26,41,71,78,103,108]. There was a trend of increased adjunctive BZD use over the years in Europe [35,80], but a slight decrease was reported in another study based in Asia [75].

BZD use was associated with older age [79], lower education levels [31], and those living in areas of low or high population density [31], greater symptom severity [79], more hostility/aggressiveness [79,85], more anxiety symptoms [31], APP or more AP use [31,102], more anticholinergic use [31], a higher number of hospitalizations in the past year [31], more EPSE [79], loss of libido [79], and mortality especially in those with chronic high-dose use [36].

## 4. Discussion

This review sought to review global real-world psychotropic prescription trends in schizophrenia and there were several main findings. Globally, there were wide inter-country and inter-regional variations in the prescription of APs, MSs, and ADs. Overall, over time there was an absolute increase in the prescription rate of SGA (increase up to 50%), MSs (increase up to 15%) and ADs (increase up to 17%), with an observed absolute decrease in the rate of APP (decrease up to 15%), use of high dose APs (decrease up to 12% in Asia), CLZ (decrease up to 9%) and LAIs (decrease up to 10%). Prescription patterns of the different agents were associated with specific socio-demographic (such as age, and gender), illness (such as illness duration, nature and severity of psychopathology, and psychiatric and medical comorbidities), and treatment factors (such as treatment setting, and adherence). 

The frequency of AP use ranged from 67% to 98.8%, instead of 100%, though APs are regarded as the primary psychotropic agent to treat schizophrenia. There are several possible explanations. Some patients with schizophrenia may be mildly ill or in remission from their illness, hence APs may not be used [36]. In addition, it is possible for studies to have incomplete data, coding errors, and reporting errors [28,39], leading to the possibility of patients not on any recorded APs. 

Our review found that the frequency of SGA use increased and more than doubled over time in three studies [43,55,73], while the prescription rate of FGA decreased. SGA is effective in ameliorating positive psychotic psychopathology, has mood stabilizing effects [112] and fewer EPSE [113], and is recommended as a first-line agent over FGA in most treatment guidelines [114]. This likely explains the notable change in AP prescription patterns with the increased use of SGA over time. The use of SGA was found to be associated with younger age and shorter duration of hospitalization which could relate to the presentation of positive symptoms at earlier presentations, and its less propensity for EPSE. However, SGAs are associated with weight gain and other metabolic side effects [115,116], thus clinicians should carefully consider the side effect profile when recommending APs to patients including SGA. 

We found a significant concern regarding treatment adherence with AP treatment, as less than half of the patients continued the treatment in several studies [41,93]. Poor AP adherence can potentially cause serious clinical consequences, such as relapse of psychotic illness, suicide attempts, and readmissions [117] and treatment discontinuation can actually paradoxically increase the total healthcare costs [118]. Although the choice of discontinuing AP agents after a period of time may be driven by concerns about safety and impact on their quality of life, it has been found that maintenance on AP drugs not only prevented relapses and readmissions but was also associated with improvements in quality of life and functioning within patients [119]. Psycho-educational interventions and medication reminders are thus important components of the holistic management approach to improve treatment adherence with psychotropic agents [120]. 

High-dose AP prescription was found to be associated with several factors, such as male gender, inpatient treatment, and more severe psychopathology. Male patients and inpatients often exhibit greater aggressive behavior and more severe psychotic symptoms, which may need higher doses of APs for clinical management together with de-escalation strategies [19]. However, high-dose AP may increase the risk of side effects, such as EPSE, including neuroleptic malignant syndrome, thus care and close monitoring during administration is needed in clinical practice [121]. 

The frequency of LAI use varied across different countries. The proportion of LAI prescriptions increased in Australia, but the use of LAI was limited in some other countries. LAIs have been associated with several clinical benefits, including better treatment adherence, reduced rates of relapse, hospitalizations, and decreased reliance on healthcare resources [122,123]. However, the proportion of LAI prescriptions is still low in many countries and possible reasons include availability and access to LAIs, high costs, especially with regard to second-generation LAIs, patient preference, clinician experience, and preference [124]. Further efforts can focus on addressing the abovementioned factors to improve the use of LAIs, and better engagement between clinicians and patients about risk–benefit discussions regarding the use of LAIs during treatment. 

We found relatively high rates of APP across countries with rates up to 70% in some European countries. Of note, APP was associated with males, inpatient treatment settings, greater number of hospitalizations, more psychiatric comorbidities, and generally longer illness duration which suggested more severe and complex illness and with likely inadequate response to AP monotherapy. APP has been linked with EPSE and other adverse effects [48,54,60,63], including weight gain [48], and excessive sedation [48] thus behooving greater attention and watchful clinical monitoring during treatment. Recent studies suggested that APP may be associated with longer periods of AP continuation [125] and reduced risk of hospitalization [126] but it is often not the first choice of treatment [127]. 

The use of CLZ varied globally with rates ranging between 19% in Asia, 22.7% in Europe to 46% in the US. Within Asia, CLZ had limited usage in Japan, but it was commonly prescribed in China [128]. CLZ has only been available in Japan since 2009 [128]. Of late, there has been a reduction in the prescription of CLZ in China, due to the stringent treatment guidelines and greater availability of alternative and new AP [19]. CLZ has been proven to be effective over other APs for treatment-resistant schizophrenia [129,130], but it also has various side effects, including myocarditis, seizure, and agranulocytosis [131]. In view of better associated adherence and lower discontinuation rate [79,91,92,94], the use of CLZ should be considered early for treatment refractory cases to allow for more expedient improvement of clinical status and functioning.

Over time, there was a general absolute rise in the prescription rates of adjunctive MS for schizophrenia patients. The use of MS was associated with numerous factors, such as longer duration of illness, APP, aggression, and suicidality, indicating more pronounced illness severity and less successful response to AP agents alone [65]. MSs, such as valproate, have been found to reduce affective symptoms [132], aggression and hostility [133], and improve global symptoms in schizophrenia [134]. In addition, lithium, has both anti-suicidal and anti-aggressive properties, making it useful for the treatment of patients experiencing these distressing symptoms [135].

Adjunctive AD usage was noted to double over time in earlier studies and has been associated with a reduction in visits to the emergency department, psychiatric hospitalizations [103], and lower mortality [36]. Several factors were linked with adjunctive AD use, including female gender, outpatient treatment setting, and previous suicidality. Patients with schizophrenia reviewed in the outpatient context may have depressive symptoms, such as low mood and suicidality, especially females, thus necessitating the initiation of ADs in combination with APs. 

### Practical Implications and Future Directions

Our review summarized extant global neuropsychopharmacological prescription trends in schizophrenia. The data help us understand the prescription trends over time, practice differences, and clinical correlates in the various geographical areas. There are several practical implications. First, by analyzing such psychotropic prescription patterns and clinical correlates, clinicians can identify patient profiles associated with specific medications and adverse effects. Second, it can raise awareness of local and global changes in the use of different psychotropic agents and prescription practices such as APP. This can increase awareness of prescription of excessive AP, and potentially lead to the simplification of the treatment regimen and reduction of side effects. Third, the prescription patterns of non-antipsychotic medications, such as ADs, MSs, and BZDs, help clinicians optimize the treatment of comorbid conditions or other symptoms among patients with schizophrenia. This can increase the overall clinical outcome and improve patient satisfaction. Lastly, the information in this review can contribute to refining psychotropic prescription practices and provide direction for future policymaking. 

However, there are several limitations in this review. First, the sample size varied widely across different studies and smaller sample sizes may affect generalizability. Second, most studies are cross-sectional, so they may not be suitable for identifying probable causality between different clinical variables and prescription patterns. Third, there is heterogeneity in clinical assessment, diagnostic methods, and data collection across the studies. Fourth, only papers published in English were included in this review. Important prescription patterns described in articles that are published in other languages might be missed. 

More research is needed in this area. Currently, the clinical correlates and outcomes associated with adjunctive medication use in schizophrenia, such as ADs and MSs, are not fully examined. More studies with larger cohorts can allow for such clinical comparisons of patients on such adjunctive medications versus not. It is also necessary to conduct cost-effectiveness analyses of different treatment options, including the long-term economic impact of various psychotropic drugs, so as to inform healthcare policies and resource allocation. Furthermore, given most current studies are cross-sectional, long-term prospective cohort studies are required to track the effects of different prescription patterns over time, including their impact on symptomatology, occurrence of side effects, and functional outcomes.

In conclusion, we found wide global variations in psychotropic prescription trends with an increase in the administration of SGA, MSs, and ADs and a decrease in the prescription rate of APP, high dose APs (up to 50% decrease in Asia), CLZ (up to 50% decrease) and LAIs. Further work, including more evidence in adjunctive neuropharmacological treatments, pharmaco-economic considerations, and examination of cohorts in longer prospective studies, can deepen understanding of changing prescription trends relevant to different treatment settings and predictors of such trends for enhancement of clinical management in schizophrenia. 

## Figures and Tables

**Figure 1 brainsci-14-00006-f001:**
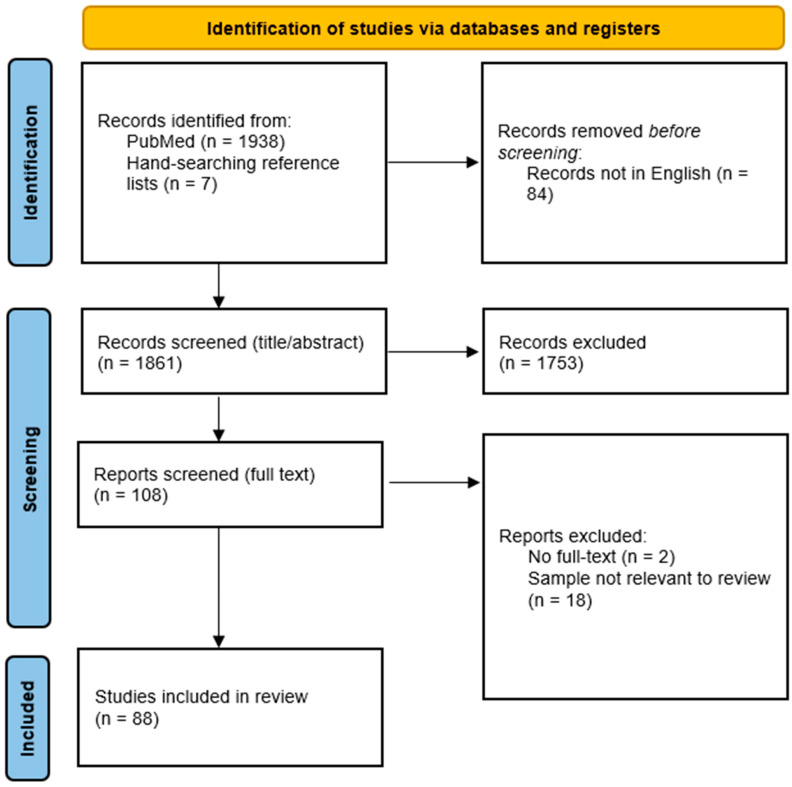
PRISMA flowchart of the study selection process.

**Table 1 brainsci-14-00006-t001:** High-dose antipsychotic prescription and clinical correlates.

	Factors Associated with Increased Frequency of High-Dose Antipsychotic Prescription	Factors Associated with Decreased Frequency of High-Dose Antipsychotic Prescription
Socio-demographic factors	Being male [19,52] Being Indigenous Australians [52]	Older age [68,90]
Treatment factors	Higher monthly cost of medications [89]	Use of second-generation antipsychotics [90]
	Antipsychotic polypharmacy [19,68,89,90]	Use of antidepressants [87]
	Use of first-generation antipsychotics [19,68]	Receiving psychotherapy [87]
	Use of long-acting injectables [68,90]	
	Use of second-generation antipsychotics [19]	
	Inpatient treatment [19]	
	Extrapyramidal, autonomic, and/or hormonal adverse effects [89]	
	Use of anticholinergics/anti-Parkinsonian drugs [19,87,89]	
	Use of anxiolytics/hypnotics [87]	
	Use of mood stabilizers [87]	
	Greater number of hospital admissions [68]	
	Longer duration of treatment [89]	
	Being under involuntary treatment [52]	
Illness factors	First-episode psychosis patients (for those on second-generation antipsychotics) [40]	Longer duration of illness (for those on first-generation antipsychotics) [40]
	Greater severity of psychopathology [19,40,68,90]	
	Current psychotic episode [89]	

**Table 2 brainsci-14-00006-t002:** Antipsychotic polypharmacy and clinical correlates.

	Factors Associated with Increased Frequency of Antipsychotic Polypharmacy	Factors Associated with Decreased Frequency of Antipsychotic Polypharmacy
Socio-demographic factors	Being Caucasian [99,102]	Being married [102]
	Being disabled [99,100,102]	Older age [27,38,45,50,99,102,104,105]
	Living in a rural area [99]	Being African-American or of a minority group [99,102]
	Living alone [104]	
	Receiving early retirement pension [104]	
	Better mental quality of life [63]	
	Being male [58,99,100]	
	Being female [104]	
Treatment factors	Prior or current clozapine use [100,104,107]	Involuntary admission [107]
	On first-generation antipsychotics [38,63,107]	Use of risperidone as principal treatment [27]
	On long-acting injectables [27,28,38,48,50,104]	Less satisfaction with treatment [63]
	On antidepressants [104]	Increased frequency of outpatient visits to psychiatry services (for users new to antipsychotic polypharmacy) [58]
	On mood stabilizers, including lithium [58,65]	Longer duration of treatment/time to treatment discontinuation [41]
	On anti-Parkinsonian drugs [58,100]	Greater number of inpatient visits [99]
	On anticholinergic agents [104]	On benzodiazepines [63]
	Use of quetiapine/paliperidone as principal treatment [27]	Poor adherence to antipsychotic treatment [100]
	Antipsychotic exposure or concomitant treatment with other psychoactive drugs [27,107]	
	Being on more classes of psychotropics [99]	
	Higher antipsychotic doses [38,48,54,104]	
	High-dose antipsychotic prescription [63,75]	
	On maintenance therapy [58]	
	Hospitalization of >30 days [75]	
	Higher number of emergency services visits [58,99]	
	Being institutionalized [104]	
	Inpatient treatment in the past year [45,102]	
	Higher self-paying cost of treatment [28]	
	Receiving electroconvulsive therapy [48]	
	Higher frequency of outpatient visits to psychiatry services [58,99,102]	
	Longer duration of treatment [51,60]	
	Medication non-adherence/poor treatment compliance [48,60]	
	Greater number of hospitalizations [50,51,58,60]	
	On benzodiazepines/anxiolytics/hypnotic drugs [58,107]	
Illness factors	Longer duration of illness [28,40,51,105]	Comorbid depression [58,102]
	More psychiatric comorbidities [45,58,100]	Comorbid anxiety [58]
	In a current episode of schizophrenia, or symptomatic [38,40,54]	Delirium [58]
	Mental retardation [45]	Higher level of functioning [54]
	Weight loss/malnutrition [100]	Older age of onset [63]
	Presence of substance use disorder/drug abuse [51,58,60]	Alcohol abuse [58,100,102]
	More medical comorbidities [99,100,104]	Presence of substance use disorder [99,102]
		More medical/physical comorbidities [102]

**Table 3 brainsci-14-00006-t003:** Mood stabilizer use and clinical correlates.

	Factors Associated with Increased Frequency of Mood Stabilizer Use	Factors Associated with Decreased Frequency of Mood Stabilizer Use
Socio-demographic factors	Being female [65,96,110]	Older age [65,69,109,110]
	Age ≥ 50 years [96]	
	Being male [88]	
	Being Caucasian [109]	
	Country/study site [65,69,79,88]	
Illness factors	Hospitalization status [65]	
	Longer duration of illness [65]	Longer duration of illness [88]
	Greater number of hospitalizations [69,88]	Hallucinations [65]
	Behavioral or verbal disorganization [65,69]	Negative symptoms [69,79]
	Aggression/hostility [65,69,79]	Cardiovascular diagnoses [109]
	Affective symptoms [65]	
	Depressive symptoms [79]	
	Social-occupational dysfunction [65]	
	Comorbid psychiatric conditions [109]	
	Previous suicidality [110]	
	Substance abuse [110]	
	Had one or more psychoses that required hospitalizations [110]	
Treatment factors	Antipsychotic polypharmacy [102]	On anti-Parkinson’s medication [109]
	On antidepressants [110]	
	On benzodiazepines [110]	
	History of antiepileptic use [109]	
	History of antipsychotic and mood stabilizer use [79]	

## Data Availability

This is a literature review of previously published records, and all records are in the public domain. Summaries of the included records are provided in Appendix A.

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
