# Peer review of "Global Neuropsychopharmacological Prescription Trends in Adults with Schizophrenia, Clinical Correlates and Implications for Practice: A Scoping Review"

_brainsci, 2023, doi:10.3390/brainsci14010006_

Round 1

Reviewer 1 Report

Comments and Suggestions for Authors

This very comprehensive review of prescription practice reveals a great deal of thought-provoking information. While it is not possible to draw conclusion about casual relationships between prescribing practice and either beneficial or harmful outcomes from a cross sectional study of this nature, it nonetheless provides some indication of factors that are not usefully examined in treatment trials with tightly defied inclusion and exclusion criteria.

However, there are several issues that the authors should address:

Exclusion of paper not in English is a serious limitation in a study of global prescription practice. While it might be impractical to repeat the review including non-English language papers, the potential problems arising for this limitation should be discussed.

Link to supplementary information did not work.

Page 3, section 3: The average number of participants (20819) per study does not sound plausible.

Page 5 The definition of high dose (‘ratio of prescribed daily dose (PDD) to defined daily dose (DDD) greater than 2.5 (PDD/DDD > 1.5),’) appears to be contradictory.

Page 6: Careless copying of text for primary sources leads to errors. E.g. the authors state:

‘Compared to those with a more continuous pattern of AP treatment, patients with poorer treatment adherence had odds of hospitalization that were 52% and 72% greater respectively [94].’’

They fail to define the two comparison groups. However, the abstract of reference 94 states:

‘Compared to individuals with a more continuous pattern of antipsychotic treatment, individuals with moderate or light use had odds of hospitalization for schizophrenia that were 52 or 72% greater (95%CI: 30-75% greater, 49-100% greater respectively).

Comments on the Quality of English Language

Careless copying from primary sources should be corrected

Reviewer 2 Report

Comments and Suggestions for Authors

While the review paper addresses an important aspect of psychotropic prescription practices in schizophrenia, there are several areas where major revisions could enhance its clarity, depth, and overall impact:

1. Specify the significance of examining psychotropic prescription practices in schizophrenia. Clearly articulate the gaps or questions in the existing literature that this review aims to address.

2. Literature Search Methodology:Provide a more detailed description of the systematic search methodology, including databases used, search terms, and inclusion/exclusion criteria. This will help readers assess the comprehensiveness of the literature review.

3. Elaborate on the wide inter-country and inter-regional variations in the prescription of psychotropic agents. Provide examples of specific countries or regions to highlight the diversity in prescription practices.

4. Offer a more nuanced discussion on the observed temporal changes in prescription rates. Specify the duration over which these changes occurred and explore potential reasons behind the observed trends.

5.Provide more in-depth insights into the socio-demographic, illness, and treatment factors influencing prescription patterns. Discuss how these factors interact and contribute to variations in psychotropic prescription practices.

6. For each category of psychotropic agents (e.g., second-generation antipsychotics, mood stabilizers, antidepressants), provide specific findings from the literature. Include key percentages, trends, and noteworthy observations to highlight the main outcomes.

7. Strengthen the discussion section by offering a more comprehensive analysis of the observed trends. Discuss potential implications for patient outcomes, healthcare resources, and future treatment approaches in schizophrenia.

8.Expand the discussion on future research directions. Propose specific areas for further investigation, such as adjunctive neuropsychopharmacological treatments, pharmaco-economic considerations, and the need for prospective cohort studies.

9. Simplify complex sentences and ensure clarity of expression. Avoid using vague terms such as "some studies" and provide specific references or examples to support statements.

10 .graphical Elements:Consider incorporating visual elements, such as tables or graphs, to present key findings more clearly. Visual aids can enhance reader comprehension and retention of information.

11/Strengthen the conclusion by summarizing the key findings and reiterating their implications for clinical management in schizophrenia. Emphasize the significance of the review in guiding future research and treatment strategies.

By addressing these aspects, the review paper can become a more robust and influential contribution to the understanding of psychotropic prescription practices in schizophrenia, providing valuable insights for clinicians, researchers, and policymakers.

Comments on the Quality of English Language

Editing required

Round 2

Reviewer 1 Report

Comments and Suggestions for Authors

THe authirs have adressed my concerns

Author Response

We thank the reviewer for the encouraging comment.

Reviewer 2 Report

Comments and Suggestions for Authors

The paper provides a valuable overview of global neuropsychopharmacological prescription patterns in adults with schizophrenia. To enhance clarity and precision, I recommend the following minor revisions:

1. Specify the importance of understanding psychotropic prescription practices in schizophrenia for changing treatment choices and patient profiles.

2. Clearly state the gap in existing research that the paper aims to address.

3. Provide examples or case studies to illustrate specific instances of prescription patterns in different regions.

4. Emphasize the clinical implications of observed trends and their potential impact on patient outcomes.

5. Suggest avenues for future research, considering adjunctive neuropsychopharmacological treatments, pharmaco-economic considerations, and prospective cohort studies.

Comments on the Quality of English Language

Fine
